# The Therapeutic Effect of Human Serum Albumin Dimer-Doxorubicin Complex against Human Pancreatic Tumors

**DOI:** 10.3390/pharmaceutics13081209

**Published:** 2021-08-05

**Authors:** Ryo Kinoshita, Yu Ishima, Victor T. G. Chuang, Hiroshi Watanabe, Taro Shimizu, Hidenori Ando, Keiichiro Okuhira, Masaki Otagiri, Tatsuhiro Ishida, Toru Maruyama

**Affiliations:** 1Department of Pharmacokinetics and Biopharmaceutics, Institute of Biomedical Sciences, Tokushima University, 1-78-1 Sho-machi, Tokushima 770-8505, Japan; kinoshita.ryo.xu@daiichisankyo.co.jp (R.K.); shimizu.tarou@tokushima-u.ac.jp (T.S.); h.ando@tokushima-u.ac.jp (H.A.); ishida@tokushima-u.ac.jp (T.I.); 2Department of Biopharmaceutics, Graduate School of Pharmaceutical Sciences, Kumamoto University, 5-1 Oe-honmachi, Kumamoto 862-0973, Japan; hnabe@kumamoto-u.ac.jp; 3Faculty of Health Sciences, Curtin Medical School, Curtin University, Perth 6845, Australia; victorchuang@hotmail.com; 4Department of Environment and Health Sciences, Osaka Medical and Pharmaceutical University, 4-20-1 Nasahara, Takatsuki 569-1094, Osaka, Japan; okuhira@gly.oups.ac.jp; 5Faculty of Pharmaceutical Sciences, Sojo University, 4-22-1 Ikeda, Kumamoto 860-0082, Japan; otagirim@ph.sojo-u.ac.jp

**Keywords:** human serum albumin, dimerization, doxorubicin, enhanced permeability and retention effect, antitumor

## Abstract

Human serum albumin (HSA) is a versatile drug carrier with active tumor targeting capacity for an antitumor drug delivery system. Nanoparticle albumin-bound (nab)-technology, such as nab-paclitaxel (Abraxane^®^), has attracted significant interest in drug delivery research. Recently, we demonstrated that HSA dimer (HSA-d) possesses a higher tumor distribution than HSA monomer (HSA-m). Therefore, HSA-d is more suitable as a drug carrier for antitumor therapy and can improve nab technology. This study investigated the efficacy of HSA-d-doxorubicin (HSA-d-DOX) as next-generation nab technology for tumor treatment. DOX conjugated to HSA-d via a tunable pH-sensitive linker for the controlled release of DOX. Lyophilization did not affect the particle size of HSA-d-DOX or the release of DOX. HSA-d-DOX showed significantly higher cytotoxicity than HSA-m-DOX in vitro. In the SUIzo Tumor-2 (SUIT2) human pancreatic tumor subcutaneous inoculation model, HSA-d-DOX could significantly inhibit tumor growth without causing serious side effects, as compared to the HSA binding DOX prodrug, which utilized endogenous HSA as a nano-drug delivery system (DDS) carrier. These results indicate that HSA-d could function as a natural solubilizer of insoluble drugs and an active targeting carrier in intractable tumors with low vascular permeability, such as pancreatic tumors. In conclusion, HSA-d can be an effective drug carrier for the antitumor drug delivery system against human pancreatic tumors.

## 1. Introduction

Pancreatic tumors remain one of the most difficult human malignancies to be treated, having the worst mortality rate and the lowest overall survival rate among all tumors. The overall survival rate for pancreatic tumors is extremely low despite rapid advances in tumor diagnosis and treatment. The prognosis of pancreatic tumors is abysmal, with 5-year survival less than 5%. Less than 10% of pancreatic tumor patients are presented with resectable disease or are suitable for potentially curative surgery. Aggressive metastasis often occurs after the operation, which is highly resistant to conventional chemotherapy and radiation therapy. Chemotherapy is the only option in metastatic pancreatic tumor treatment, but sadly, most of the time, chemotherapy is purely palliative. Despite gemcitabine being the standard first-line treatment, gemcitabine-based combination chemotherapy showed either marginal or no improvement in survival for advanced pancreatic tumors. Pancreatic tumor patients with locally advanced disease have 6–10 months of median survival, and patients with metastatic disease only have 3–6 months of median survival. Hence, novel strategies to treat pancreatic tumors are urgently needed.

Effective drug delivery in pancreatic tumor treatment remains a major challenge. Improving the specificity and stability of the delivered chemotherapeutic agent using ligand or antibody-directed delivery represents a significant problem. Recent advances in antibody-drug conjugate (ADC) technology have led to the development of cancer-stroma targeting (CAST) therapy as a novel antitumor drug delivery system, especially for refractory, stromal-rich tumors, such as pancreatic tumors [1]. This CAST strategy is developed based on the aggressiveness of the tumor; the more aggressive, the greater the deposition of insoluble fibrin in tumor tissue. Peptide–drug conjugates (PDC) have recently gained significant attention as tools for developing specific delivery systems for pancreatic tumors [2]. Using a human pancreatic ductal adenocarcinoma cell line, PDC was sufficiently incorporated by the cells within 2 h, whereas ADC was not visible in the cell after 2 h. In addition, an in vivo study using tumor-bearing mice indicated that PDCs incorporating this peptide might exert potent antitumor effects without missing the target cells. PDC was efficiently and selectively incorporated into target pancreatic tumor lesions, including primary and metastatic sites. Conversely, current ADCs, such as anti-HER2 and anti-EGFR antibodies, are limited to targeting specific antigen-positive tumor cells. 

Doxorubicin (DOX) is an anthracycline antibiotic used as a chemotherapeutic agent to treat a wide variety of tumors, including lymphoma, lung tumor, stomach tumor, breast tumor, and osteosarcoma [3,4]. Cardiotoxicity is a major clinical adverse reaction of DOX upon cumulative dosing. Liposome-based drug delivery systems are known to facilitate targeting of specific tumor treatment agents, improve pharmacokinetics, reduce side effects, and potentially increase tumor uptake for pancreatic tumor therapy. Liposomal DOX has been reported to alleviate cardiotoxicity with improved antitumor activity. In particular, pegylated liposomal DOX has shown significant pharmacologic advantages and an added clinical value over DOX. Doxil^®^, a PEGylated liposome preparation for passive targeting aimed at reducing the side effects of DOX, and various DOX-loaded nano-drug delivery system (DDS) preparations [5,6], including the micellar preparation NK911, have been developed [7].

Human serum albumin (HSA) has gained popularity as a nano-drug delivery carrier. Aldoxorubicin (INNO-206), an HSA binding prodrug of DOX, has been developed to increase tumor targeting efficiency. INNO-206 binds to the SH group of Cys-34 of endogenous HSA via the maleimide group in the linker structure attached to DOX [8,9]. Consequently, HSA-conjugated INNO-206 exhibits improved blood retention compared with free DOX. Recent clinical trials examined the efficacy of INNO-206 in treating soft tissue sarcoma and AIDS-related Kaposi’s sarcoma and preclinical trials for solid tumors, such as pancreatic tumors [10]. Administered at the equivalent dose, INNO-206 is safer than free DOX due to its improved kinetics and antitumor activity [9]. This evidence highlights the need for further attempts to increase DOX potency, exploring the potential of a novel drug carrier with higher drug accumulation in pancreatic tumors. 

The current nanoparticle albumin-bound (nab)-technology, such as nab-paclitaxel, uses HSA as a natural solubilizer of insoluble drugs, not as a tumor-targeting carrier [11]. After intravenous administration, nab-paclitaxel (Abraxane^®^) dispersed rapidly and behaved similarly to HSA monomer in the general circulation. Consequently, nab-paclitaxel almost lost its tumor-targeting ability via the enhanced permeability and retention (EPR) effect [12]. Kim’s group developed anti-PD-L1 antibody conjugated nab technology as a novel combination of chemotherapeutic and immune-therapeutic antitumor approaches to improve its tumor-targeting ability. This approach demonstrated that the pH-responsive drug release and PD-L1 targeting does enhance the tumor selectivity [13]. Furthermore, insulin-like growth factor 1 receptor inhibitors enhance uptake and efficacy of nab-PTX by mimicking glucose deprivation and promoting macropinocytosis via AMPK, a nutrient sensor in cells. This suggests that nanoparticulate albumin-bound drug efficacy can be therapeutically improved by reprogramming nutrient signaling and enhancing macropinocytosis in tumor cells [14].

Previously, we demonstrated that HSA dimers (HSA-d) have higher blood circulation activity and lower vascular permeability than HSA monomers (HSA-m) [15]. Interestingly, HSA-d possesses a higher tumor distribution than HSA-m [16,17], so *S*-nitrosated HSA-d could enhance the EPR effect via an endogenous albumin transport (EAT) system [18]. Tumor cells actively “EAT” (consume) endogenous HSA as a source of amino acids to continue cell proliferation via several HSA receptors, such as gp60 or SPARC [19]. In particular, the EAT system should be activated in order to survive in a hypoxic region with inferior blood vessel density [20]. Therefore, EAT system targeting is a promising DDS strategy using HSA-d for refractory tumor therapy.

Herein, we designed the HSA-d-doxorubicin (HSA-d-DOX) as a next-generation nab technology therapeutic agent with both properties: as a natural solubilizer of insoluble drugs and active targeting capacity. DOX was conjugated to HSA-d via a tunable pH-sensitive linker for the controlled release of DOX. pH-responsive hydrazone bonds between DOX and the linker maintain the binding in normal tissues and blood circulation (pH 7.4–7.6). The bond is cleaved only in an acidic environment, such as tumor tissues, so it becomes possible to release DOX selectively. The antitumor activity of HSA-d-DOX was evaluated in vitro and in vivo using the SUIzo Tumor-2 (SUIT2) human pancreatic tumor model as a refractory tumor.

## 2. Materials and Methods

### 2.1. Materials

DOX was purchased from Wako Pure Chemical Industries, Ltd. (Osaka, Japan). 2-Iminothiolane hydrochloride and Amicon Ultra (4 and 15) 10 kDa were purchased from Merck KGaA (Darmstadt, Germany). INNO-206 was purchased from ChemScene (Monmouth Junction, NJ, USA). Cell Counting Kit-8 (CCK-8) was obtained from Dojindo Molecular Technologies (Kumamoto, Japan). Other chemicals were of the best commercially available grades, and all solutions were made using deionized water.

### 2.2. Expression and Purification of HSA-m and HSA-d

Recombinant HSA-m and HSA-d were expressed by *Pichia pastoris* (*P. pastoris*) [21] and defatted [22]. Briefly, these constructed plasmids (pPIC9-HSA-m and pPIC9-HSA-d) were transferred to XL10-Gold *Escherichia coli* (*E. coli*). *P. pastoris* GS115 his4 was transformed with SalI-digested pPIC9-HSA-m or pPIC9-HSA-d by electroporation. The protocol used to express the HSAs was a modification of a previously published protocol [23]. Single colonies of *P. pastoris* were grown for 48 h at 30 °C in buffered minimal glycerol-complex medium until an A_600_ value of 2–4 was obtained. Cells were then harvested by centrifugation at 3000× *g*, and cell pellets were washed extensively and resuspended in buffered minimal methanol-complex medium to an approximate A_600_ value of 10–15. Then, the baffled flasks were shaken for 96 h at 30 °C, with a daily addition of methanol at a final concentration of 1% to maintain the induction conditions. The recombinant HSAs were purified after 96 h of induction [24]. Preparation of the HSAs was first subjected to chromatography with the Blue Sepharose 6 Fast Flow column (Cytiva, Tokyo, Japan) equilibrated with 200 mM sodium acetate buffer (pH 5.5) after dialysis with the same buffer. The eluted HSAs were deionized and defatted via charcoal treatment, freeze-dried, and then stored at −80 °C until used.

### 2.3. Cells and Animals

Human pancreatic tumor transferred luciferase gene SUIzo Tumor-2 (SUIT2) cells were cultured in DMEM + 10% fetal bovine serum with antibiotics (100 units penicillin/mL, and 100 μg streptomycin/mL). The cells were passaged when approximately 90% confluence was reached. BALB/c nu/nu mice (male, 5 to 6 weeks old, Japan SLC Inc., Shizuoka, Japan) was used as SUIT2-bearing mice. SUIT2 bearing mice were prepared by subcutaneous transplantation with 1 × 10^6^ cells into the back of the mice [17]. All animal experiments were carried out according to the Laboratory Protocol for Animal Handling T2019-47 (1 August 2019) of Tokushima University.

### 2.4. Preparation of HSA-DOX

HSA-m or HSA-d (150 μM) in 100 mM KPB + 0.5 mM DTPA (pH 7.8) incubated with 2-iminothiolane (final concentration 3.6 mM) and mixed gently for 1 h at 25 °C [25]. INNO-206 (final concentration 1.8 mM) was added to the solution under dark conditions and mixed at 25 °C for 3 h. Then, the unreacted 2-iminothiolane and INNO-206 were removed by ultrafiltration using Amicon Ultra 4 (NMWL: 10 kDa). Next, HSA-m-DOX or HSA-d-DOX was dialyzed against deionized water and lyophilized. These samples were stored at −20 °C until used. The particle sizes and polydispersity index (PDI) of HSA-DOX under PBS (pH 7.4) condition were recorded using a Malvern zetasizer Nano ZS (Malvern Instruments, Worcestershire, UK).

### 2.5. Quantification of DOX Loaded to HSA

To quantify DOX loaded to HSA, the absorbance (490 nm) of HSA-DOX was measured using 96-well plate by the iMark microplate reader (Bio-Rad, Hercules, CA, USA). Free DOX solutions (6.25, 12.5, 25, 50 μg/mL) were used as standard. The same protein concentration without DOX was also measured to adjust for background absorption. The protein concentration of HSA-DOX was determined by the Bradford method. DOX loading efficiency (DOX/HSA) was calculated using this concentration of HSA.

### 2.6. In Vitro Release Profile of DOX from HSA-m-DOX or HSA-d-DOX

To evaluate the stability of HSA-m-DOX or HSA-d-DOX in acidic and neutral pH conditions, HSA-m-DOX or HSA-d-DOX was dissolved with PBS (pH 7.4) and acetic acid buffer (pH 5.5), respectively, and adjusted to 2.0 μg (DOX)/mL. Then, HSA-m-DOX or HSA-d-DOX was incubated at 37 °C in each pH condition, and fluorescence intensity (Ex/Em = 488 nm/585 nm) was measured using a spectrofluorimeter FP-8200ST (JASCO) at 0, 3, 6, 12, 24, 36, 48 h. The % release of DOX was calculated as follows.
% release of DOX = (A − B)/(C − B) × 100

A: Fluorescence intensity of sample;

B: Fluorescence intensity at 0% release condition (pH 7.4, 25 °C);

C: Fluorescence intensity at 100% release condition (pH 1.0, 37 °C, 2 h).

### 2.7. In Vitro Antitumor Activity of HSA-DOX

SUIT2 cells (1 × 10^4^ cells/well) were seeded in a 96-well plate. After being left overnight, free-DOX, HSA-m-DOX or HSA-d-DOX (10–5000 ng (DOX)/mL) was added to the cells and reacted for 48 h. HSA-m and HSA-d were used as a negative control. Cell viability was assessed by CCK-8.

### 2.8. Quantification of Intracellular DOX 

SUIT2 cells (1 × 10^5^ cells/well) were seeded in a 12-well plate. After being left overnight, the medium was replaced with a serum-free medium, then incubated for 2 h. After serum starvation, free-DOX, HSA-m-DOX or HSA-d-DOX (10 nmol (DOX)/mL) was added, and the cells were harvested 2 h later. The collected cell suspension was centrifuged (1500× *g*, 5 min, 4 °C), and the supernatant was removed. Then, 2 N HCl (500 μL) was added to the cell pellet to lyse the cells. Subsequently, deproteinization was performed by adding methanol (500 μL) and centrifugation (1500× *g*, 5 min, 4 °C), and the fluorescence intensity of the supernatant (Ex/Em = 488 nm/585 nm) was spectrophotometrically determined using FP-8200ST (JASCO).

### 2.9. Pharmacokinetic Analysis of HSA-DOX

INNO-206, HSA-m-DOX or HSA-d-DOX (8.0 mg (DOX)/kg) was intravenously administered in SUIT2 human pancreatic tumors and subcutaneously implanted in mice with a tumor size of 100 mm^3^. Then, some main organs (heart, lung, liver, spleen, and kidney) and the tumor were extirpated at 6 h after administration, and ex vivo imaging was performed using an IVIS imaging system. The DOX fluorescence intensity of these tissues was measured at Ex/Em = 465 nm/600 nm.

### 2.10. In Vivo Antitumor Activity and Side Effects of HSA-DOX

When tumors reached 100 mm^3^ in mice implanted subcutaneously with SUIT2 human pancreatic tumor, the mice were divided into cohorts (*n* = 4) and treated intravenously with PBS (control), INNO-206 (8.0 mg (DOX)/kg), HSA-m-DOX (8.0 mg (DOX)/kg), or HSA-d-DOX (8.0 mg (DOX)/kg) on days 0 and 7, and then monitored for 21 days [17]. During treatment, tumor volume and body weight were measured daily, and blood was collected from the inferior vena cava on day 21 to measure various biochemical parameters. Specifically, after measuring blood cell parameters using a multi-item automatic blood cell counter KX-21N (Sysmex, Kobe, Japan), centrifugation (1500× *g*, 10 min) was performed. After the collection of serum, liver injury markers (AST, ALT) and kidney injury markers (BUN) were evaluated using the respective measurement kit (Wako Pure Chemical Industries, Ltd., Tokyo, Japan).

### 2.11. Statistical Analysis 

The experimental data are presented as the means ± standard deviation. The statistical significance of differences between groups was analyzed with Student’s *t*-test or ANOVA with Tukey’s post hoc test. A probability value of *p* < 0.05 was considered to indicate statistical significance.

## 3. Results and Discussion

### 3.1. Preparation of HSA-d-DOX

First, HSA-m-DOX and HSA-d-DOX were synthesized using the method shown in Figure 1A. Specifically, the primary amine of the HSA molecule is chemically modified with an SH group using iminothiolane. Then, the maleimide group was reacted beforehand with the pH-responsive linker of INNO-206. As a result, the particle sizes of HSA-m-DOX and HSA-d-DOX were 3.98 nm and 7.52 nm, and amounts of DOX binding to HSA-m-DOX and HSA-d-DOX were 4.08 mol/mol HSA-m and 7.9 mol/mol HSA-d, respectively (Table 1). These results suggest that HSA-m-DOX and HSA-d-DOX exist as almost homogeneous molecules without aggregated proteins. In the previous report, *N*-succinimidyl *S*-acetylthioacetate (SATA) reagent was used to introduce SH to amino groups [26]. In the case of the SATA reagent, the terminal is thioester after reaction, so a deprotection procedure with hydroxylamine is needed. In contrast, the preparation process was successfully shortened by one step using iminothiolane, where a deprotection operation could be omitted. Fortunately, the DOX loading efficiency using iminothiolane was equivalent to the value in the reference using SATA [26]. The stability of HSA-m-DOX and HSA-d-DOX after lyophilization was assessed by evaluating the particle sizes and DOX-loading rates. These data showed that both HSA-m-DOX and HSA-d-DOX were suspendable after lyophilization, and no other change, such as aggregates, can be observed (Figure 1B). In addition, since the DOX-loading rates of both compounds before and after lyophilization were the same, this result indicated that storage of HSA-m-DOX and HSA-d-DOX as a lyophilisate was possible (Figure 1C). 

### 3.2. In Vitro Release Profile of DOX from HSA-m-DOX or HSA-d-DOX

Although DOX is a fluorescent substance, fluorescence self-quenching occurs when DOX is bound to HSA-m and HSA-d. The DOX release profile of HSA-m-DOX or HSA-d-DOX in buffers in different pH conditions was evaluated using this fluorescence self-quenching phenomenon. The release of DOX from HSA-m-DOX or HSA-d-DOX was not observed in the pH 7.4 buffer. In contrast, acidic conditions (pH 5.5) accelerated the release of DOX (Figure 2). These data suggest that the DOX release from HSA-m-DOX and HSA-d-DOX occurs in acidic conditions, like those in tumor tissues. In tumor tissues, malnutrition and hypoxia result from incomplete blood vessel construction and uncontrolled growth of the tumor cells. The release profile of HSA-m-DOX was very similar to HSA-d-DOX. The total release from HSA-d-DOX in pH 5.5 buffer at 48 h was around 80%, suggesting a result that is similar when compared with another published release assay for DOX [27]. As a result, glycolytic metabolism is enhanced, leading to the buildup of an acidic environment [28]. In particular, the acidic environment is more pronounced in pancreatic tumors. Therefore, after reaching the tumor tissue via the EPR effect, HSA-d-DOX is expected to release DOX more efficiently in pancreatic tumors.

### 3.3. In Vitro Antitumor Activity of HSA-d-DOX against Human Pancreatic Tumor Cells

To confirm the antitumor activity of HSA-d-DOX on SUIT2 human pancreatic tumor cells, an evaluation was performed. Free-DOX, HSA-m-DOX, and HSA-d-DOX were incubated with SUIT2 cells for 48 h, and the number of viable cells were quantified by CCK-8 assay. Figure 3A shows that free-DOX possessed the highest cytotoxicity against SUIT2 cells among all drugs, and HSA-d-DOX has significantly inhibited the survival rate compared with HSA-m-DOX (Figure 3A, Table 2). In addition, carriers alone, such as HSA-m or HSA-d, did not show any cytotoxicity (Table 2). To clarify the cytotoxic mechanism of HSA-d-DOX, the amount of intracellular DOX was analyzed. Free-DOX, HSA-m-DOX or HSA-d-DOX was incubated with SUIT2 cells for 2 h, and the amount of intracellular DOX was quantified by fluorescent spectroscopy. Figure 3B shows the amount of intracellular DOX in decreasing order, free-DOX > HSA-d-DOX > HSA-m-DOX. This result was highly consistent with the cytotoxicity results of these compounds. Previously, we reported that HSA-d-DOX is also taken into cells via caveolin-1-mediated macropinocytosis [17,29]. Additionally, pancreatic tumor cells are known to have increased expression of secreted protein acidic and are rich in cysteine (SPARC) for the EAT system [20]. This EAT System is activated at a region where the EPR effect must be enhanced, which is therefore considered a region where it is traditionally difficult to deliver a polymeric antitumor agent [18]. 

### 3.4. Biodistribution of HSA-d-DOX

In order to confirm the tumor accumulation of HSA-d-DOX in tumor-bearing mice, the organ distribution of HSA-d-DOX in vivo was evaluated using a SUIT2 human pancreatic tumor subcutaneous transplantation model. INNO-206, HSA-m-DOX or HSA-d-DOX was intravenously administered in the SUIT2 subcutaneous transplantation model at a dose of 8 mg (DOX)/kg. The fluorescence intensity of DOX in each organ was measured by ex vivo imaging at 6 h after administration (Figure 4A). Intriguingly, HSA-d-DOX showed significantly higher tumor accumulation of DOX than other groups (Figure 4B). In addition, the liver distributions of HSA-m-DOX and HSA-d-DOX were higher than INNO-206, strongly suggesting that liver injury markers, such as AST and ALT, should be measured after the administration of HSA-m-DOX or HSA-d-DOX to clarify whether these administrations induce side effects related to liver injury. Previously, our collaborators demonstrated that modifications of basic amino acids on the surface of HSA have increased liver uptake of the HSA by 30-fold [30,31]. In general, modifications of basic amino acids led to the blockage of the positive charges of HSA. To conjugate DOX to HSA in this study, 2-iminothiolane reacted with lysine residues on the surface of HSA. This reaction induced the blockage of the positive charges. Therefore, this evidence suggested that this 2-iminothiolane reaction may induce the increase of liver uptake of HSA-m-DOX and HSA-d-DOX. To avoid this problem, the other method of DOX conjugation without the modification of basic amino acids would be necessary.

### 3.5. In Vivo Antitumor Activity of HSA-d-DOX

Finally, the therapeutic effect of HSA-d-DOX was evaluated using a SUIT2 human pancreatic tumor subcutaneous transplantation model to verify whether HSA-d-DOX exerts an antitumor effect in the SUIT2 human pancreatic tumor implantation model, which shows resistance to macromolecular antitumor drugs. SUIT2 tumor-bearing mice with a tumor volume of about 100 mm^3^ were divided into Control, INNO-206, HSA-m-DOX, and HSA-d-DOX administration groups. The dose of INNO-206, HSA-m-DOX, and HSA-d-DOX was adjusted to 8.0 mg (DOX)/kg, which was intravenously administered to SUIT2 tumor-bearing mice on days 0 and 7. The data showed that HSA-d-DOX significantly suppressed tumor growth compared to INNO-206 and HSA-m-DOX (Figure 5A). Regarding body weight changes, although each group of INNO-206, HSA-m-DOX or HSA-d-DOX showed a slight weight loss after the second administration, there was no significant change overall (Figure 5B). 

The following dosing rates have been reported to be effective: on day 10, 17, and 24, dosings of DOX 24 mg/kg [9], and on day 10, 17, and 24, dosings of DOX 12 mg/kg [32], while the dosing rate used in the present study, on day 0 and 7, dosings of DOX 8.0 mg/kg, practically showed no antitumor efficacy. This 2 × DOX 8.0 mg/kg dosing rate explains why the antitumor activity of INNO-206 did not show a significant difference from that of the PBS treated group. In contrast, despite dosing at 2 × DOX 8.0 mg/kg, HSA-d-DOX showed antitumor activity. 

To evaluate the onset of side effects, various biochemical parameters were measured at day 21. These biochemical parameters, including AST and ALT, showed no significant difference among all groups (Table 3). These data indicate that HSA-d-DOX possesses the highest therapeutic effect against human pancreatic tumors without severe side effects. Previously, we examined the antitumor effect of Abraxane^®^ using the same SUIT2 human pancreatic tumor implantation model [17]. It is well-known that this model shows higher resistance to most antitumor drugs compared with C26 murine colon tumor and B16 murine melanoma subcutaneous inoculation models. The administration of Abraxane^®^ (20 mg (paclitaxel)/kg) could not significantly inhibit the tumor growth like INNO-206. Although Table 2 shows that free DOX possesses higher cytotoxicity than HSA-d-DOX in vitro, HSA-d-DOX has the highest antitumor activity among other drugs. These data strongly suggest that the tumor accumulation and retention of HSA-d-DOX are significantly superior to other drugs. Taken together, HSA-d-DOX has the potential to be a promising macromolecular antitumor drug. 

## 4. Conclusions

In this study, HSA-d-DOX was developed as a novel antitumor drug nano-DDS loaded with DOX using next-generation nab technology. HSA-d exhibited excellent tumor migration and cellular uptake ability compared with HSA-m using conventional nab technology. HSA-d-DOX efficiently delivered DOX to pancreatic tumor cells, inducing a potent antitumor effect without severe side effects. Therefore, dimerization of HSA could function not only as a natural solubilizer of insoluble drugs, but also as the active targeting carrier in low vascular permeability or an intractable pancreatic tumor.

## Figures and Tables

**Figure 1 pharmaceutics-13-01209-f001:**
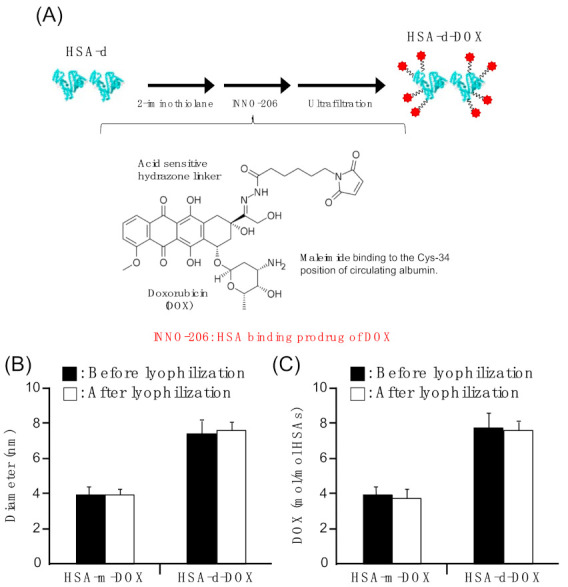
The preparation scheme of HSA-d-DOX, and the stability of HSA-m-DOX and HSA-d-DOX particles before and after lyophilization. (**A**) Preparation scheme of HSA-d-DOX. (**B**) The change of particle size and (**C**) DOX loading of HSA-DOX after lyophilization of HSA-m-DOX and HSA-d-DOX. Data are averages ± standard deviation (*n* = 4).

**Figure 2 pharmaceutics-13-01209-f002:**
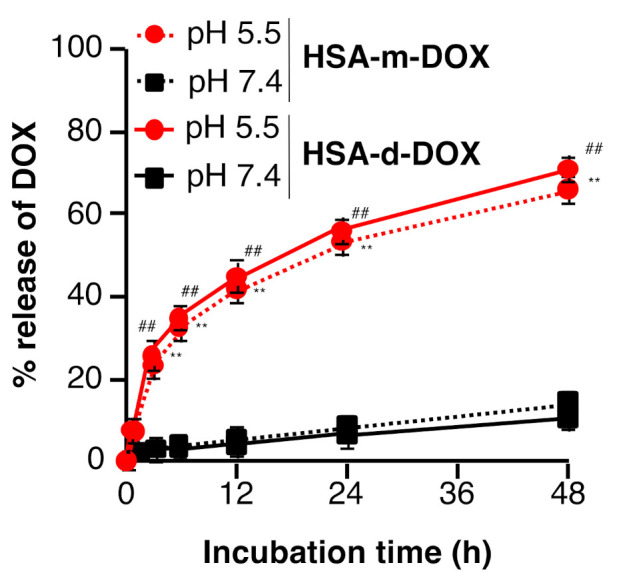
In vitro pH-dependent DOX release profile of HSA-m-DOX or HSA-d-DOX. HSA-m-DOX or HSA-d-DOX was diluted with PBS (pH 7.4) or acetate buffer (pH 5.5) to 2.0 mg (DOX)/mL. After incubation, the fluorescence intensity (Ex/Em: 488 nm/585 nm) was measured. Data are averages ± standard deviation (*n* = 4). ** *p* < 0.01 vs. HSA-m-DOX in pH 7.4 group. ^##^
*p* < 0.01 vs. HSA-d-DOX in pH 7.4 group.

**Figure 3 pharmaceutics-13-01209-f003:**
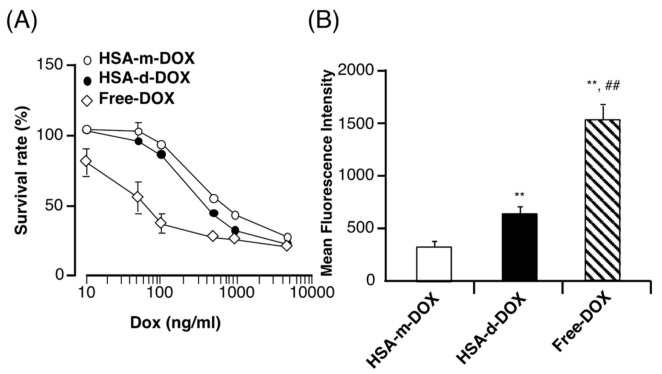
Cytotoxicity of HSA-d-DOX on pancreatic tumor cell. (**A**) SUIT-2 cells were treated with various concentration of HSA-m-DOX, HSA-d-DOX or free-DOX for 48 h. Survival cells were quantified by CCK-8 assay and expressed as percent survival to untreated control. (**B**) Cellular uptake of HSA-d-DOX by pancreatic tumor cells. SUIT2 cells were treated with HSA-m-DOX, HSA-d-DOX, free-DOX (150 μg/mL) in serum-free DMEM. Cellular uptake of DOX was analyzed by fluorescent spectroscopy (Ex/Em: 488 nm/585 nm; Data are averages ± standard deviation (*n* = 6). ** *p* < 0.01 vs. HSA-m-DOX, ^##^
*p* < 0.01 vs. HSA-d-DOX.

**Figure 4 pharmaceutics-13-01209-f004:**
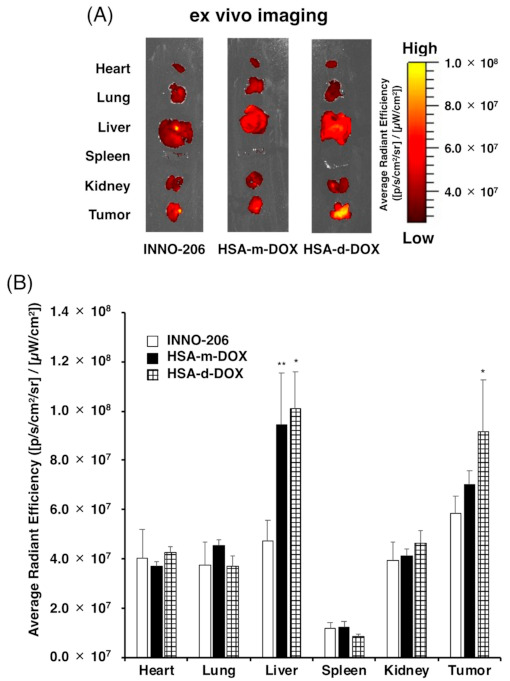
Organ distribution of HSA-DOX. (**A**) ex vivo imaging, (**B**) ex vivo radiant efficiency of the major organs from SUIT2 ectopic tumor bearing mice at 6 h after 8.0 mg (DOX)/kg of INNO-206, HSA-m-DOX and HSA-d-DOX were IV injected. The results obtained for the control group were used to correct for nonspecific background fluorescence. Data are averages ± standard deviation (*n* = 3). * *p* < 0.05, ** *p* < 0.01 vs. INNO-206.

**Figure 5 pharmaceutics-13-01209-f005:**
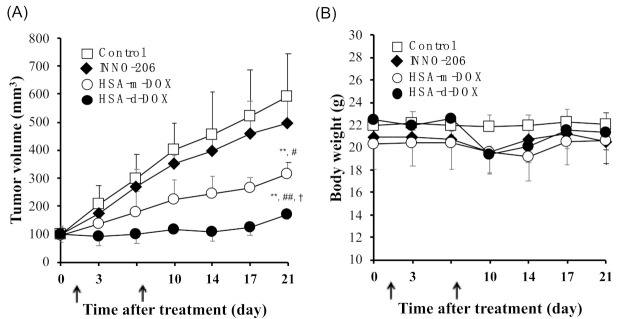
The antitumor activity of HSA-d-DOX in SUIT2 ectopic tumor bearing mice. (**A**) The tumor volume and (**B**) body weight were measured in SUIT2 ectopic tumor bearing mice at each of the selected time points. The mice were IV injected with PBS, INNO-206 (8.0 mg (DOX)/kg), HSA-m-DOX (8.0 mg (DOX)/kg) or HSA-d-DOX (8.0 mg (DOX)/kg) at each of the selected time points. The arrows indicate days of treatment. Data are averages ± standard deviation (*n* = 4). ** *p* < 0.01 vs. control, ^#^
*p* < 0.05, ^##^
*p* < 0.01 vs. INNO-206, ^†^
*p* < 0.05 vs. HSA-m-DOX.

**Table 1 pharmaceutics-13-01209-t001:** Physicochemical characterization of HSA-m-DOX and HSA-d-DOX.

	Diameter (nm)	PDI	DOX/HSAs (Molar Ratio)
HSA-m-DOX	3.98 ± 0.5	0.426 ± 0.01	4.08 ± 0.26
HSA-d-DOX	7.52 ± 0.7	0.297 ± 0.01	7.90± 0.46

Data are averages ± standard deviation (*n* = 4).

**Table 2 pharmaceutics-13-01209-t002:** IC_50_ of DOX derives against pancreatic tumor cell.

Treatment Groups	IC_50_ (DOX ng/mL)
HSA-m	-
HSA-d	-
HSA-m-DOX	676.7 ± 52.4
HSA-d-DOX	454.2 ± 66.7 *
free-DOX	62.21 ± 24.5 **

Data are averages ± standard deviation (*n* = 6). * *p* < 0.05, ** *p* < 0.01 vs. HSA-m-DOX.

**Table 3 pharmaceutics-13-01209-t003:** Blood chemistry parameters after drugs treatment.

	WBC (×10^2^/μL)	RBC (×10^4^/μL)	HGB (g/dL)	PLT (×10^4^/μL)	AST (IU/L)	ALT (IU/L)	BUN (mg/dL)
**Control**	24.3 ± 3.06	944.0 ± 32.5	114.4 ± 18.3	64.1 ± 12.4	19.6 ± 4.5	29.9 ± 3.2	78.8 ± 8.5
**INNO-206**	20.5 ± 5.9	924.8 ± 47.8	106.5 ± 17.5	57.7 ± 11.1	16.6 ± 5.1	39.5 ± 4.4	78.3 ± 10.4
**HSA-m-DOX**	26.3 ± 8.3	934.8 ± 88.5	115.1 ± 13.2	62.8 ± 2.5	17.2 ± 5.1	33.7 ± 2.4	82.5 ± 19.3
**HSA-d-DOX**	17.8 ± 3.1	939.8 ± 79.0	102.6 ± 30.7	66.1 ± 7.5	20.4 ± 3.9	37.7 ± 2.6	78.8 ± 14.3

Parameters of blood cells (white blood cell (WBC) count, red blood cell (RBC) count, hemoglobin (HGB) count, platelet (PLT) count), aspartate transaminase (AST), alanine aminotransferase (ALT), blood urea nitrogen (BUN) in C26 ectopic tumor bearing mice at day 28. Data are averages ± standard deviation. (*n* = 4).

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
