# Peer review of "The Therapeutic Effect of Human Serum Albumin Dimer-Doxorubicin Complex against Human Pancreatic Tumors"

_pharmaceutics, 2021, doi:10.3390/pharmaceutics13081209_

Round 1

Reviewer 1 Report

The manuscript “The therapeutic effect of HSA dimer-doxorubicin comple against human pancreatic tumour” by Kinoshita et al. describes the characterization of and application of a drug delivery system for the treatment of pancreatic cancer.

While the topic is of interest and the manuscript follows a distinct outline, I would not recommend publication in its current state since there are multiple minor but some major flaws.

Major issues:

I do no want to critisize a manuscript due to being short but here some major sections are not detailed enough. The introduction lacks a state of the art for pancreas treatment with drug delivery systems. Only the use of HAS without considering other systems is described here.

Please add a thorough description of the state of the art.

The next section (Material and Methods) is my major concern. Here the methods are not well described and therefore cannot be reproduced. This starts with the expression conditions of HAS in Pichia pastoris followed by the conditions of cell cultures (concentration of antibiotics is missing, cultivation time and volume is missing).

A particle size measuring is described but without mentioning the details. The particle sizes cannot be found in the results section. Can you add them?

The results and discussion part lacks a discussion. Only results are described but not discussed with literature. There is no discussion with other treatment methods.

Minor flaws:

Latin words such as ex vivo and in vitro as well as the names of yeast cells such as Pichia pastoris should be written italic.

There are multiple errors with numbers and units (some units are directly after the number and sometimes it seems they are separated by two spaces).

Some exponents are not written in superscript such as mm3

Tables and figures are not correctly embedded in the text (but this might be an editorial issue).

Figure 1 demonstrates small labellings. Either the labellings or the whole scheme should be increased.

Author Response

Major issues:

(Q) I do no want to critisize a manuscript due to being short but here some major sections are not detailed enough. The introduction lacks a state of the art for pancreas treatment with drug delivery systems. Only the use of HSA without considering other systems is described here.

Please add a thorough description of the state of the art.

<Reply>

Thank you very much for your constructive comments. We added some information regarding other systems using antibody-drug conjugate (ADC) and peptide-drug conjugates (PDC) technologies. We have added them to the revised manuscript (page2).

(Q) The next section (Material and Methods) is my major concern. Here the methods are not well described and therefore cannot be reproduced. This starts with the expression conditions of HSA in Pichia pastoris followed by the conditions of cell cultures (concentration of antibiotics is missing, cultivation time and volume is missing).

<Reply>

Thank you very much for your kind comments. We added the methods for Expression and purification of HSA-m and HSA-d, and cell culture condition. We have added them to the revised manuscript (page3).

(Q) A particle size measuring is described but without mentioning the details. The particle sizes cannot be found in the results section. Can you add them?

<Reply>

We added the condition of particle sizes measurement and the result and discussion in the revised manuscript (pages 4 and 5)

(Q) The results and discussion part lacks a discussion. Only results are described but not discussed with literature. There is no discussion with other treatment methods.

 <Reply>

Thank you very much for your constructive comments. We previously demonstrated an antitumour effect of Abraxane on the same SUIT-2-bearing mice model. So, we have described some discussion compared with the literature in the revised manuscript (page 10).

Minor flaws:

(Q) Latin words such as ex vivo and in vitro as well as the names of yeast cells such as Pichia pastoris should be written italic.

<Reply>

We had corrected it.

(Q) There are multiple errors with numbers and units (some units are directly after the number and sometimes it seems they are separated by two spaces).

<Reply>

We had corrected them.

(Q) Some exponents are not written in superscript such as mm3

<Reply>

We had corrected it.

(Q) Tables and figures are not correctly embedded in the text (but this might be an editorial issue).

<Reply>

We had re-attached all tables and figures.

Figure 1 demonstrates small labellings. Either the labellings or the whole scheme should be increased.

<Reply>

We had corrected it.

Reviewer 2 Report

This manuscript reports the results on the therapeutic effect of HSA dimer-doxorubicin complex against human pancreatic tumor.

The manuscript described the preparation and evaluation of HSA dimer (HSA-d)-doxorubicin conjugate. The authors demonstrated that the conjugate inhibited the tumor growth in tumor-bearing mice without serious side effects. Overall, the study was appropriately designed and investigated. However, there are some issues that require clarifications. Also, the manuscript requires intensive language correction. Please consider the below comments.

  1. The Introduction section: please provide more information on nanoparticle albumin-bound (nab) with recently published references.
  2. Section 2.3. Preparation of HSA-DOX: please rewrite all the section since it was unclear. Please describe the lyophilization process in detail.
  3. There was no statistical analysis. Please include in the revised manuscript.
  4. Section 3.2. In vitro release profile of DOX from HSA-d-DOX (Fig 2): the release of free-DOX in two pH conditions should be determined for the comparison with the formulation group.
  5. The x-axis of the Figure 3A should be log-scaled.
  6. In the Figure 4B, the reviewer found that the liver tissue has higher accumulation and HSA-d-DOX seems to have the highest values compared to control and HSA-m-DOX group. Please discuss this result somewhere in terms of active targeting. The high accumulation of the liver does not indicate the active targeting of the formulation. Please clarify it.
  7. What is the loading efficiency of the DOX for HSA-m-DOX and HSA-d-DOX. Is there any advantage of HSA-d-DOX?

Minor points:

  1. Title: please consider avoiding abbreviation (HAS).
  2. Please define all the abbreviations before use (e.g., DDS, SUIT2 in line 74).
  3. Please correct the units in lines 105, 113, and 133.
  4. Lines 150-151: HSA-m-DOX (8.0 mg (DOX)/kg) was repeated. Please check again.
  5. Table 2: *P < 0.05 vs. HSA-m-DOX. Free-DOX and HSA-m-DOX were significantly different but there was no significant symbol.
  6. Lines 349-352: please check.
  7. Please carefully check the English usage for the whole manuscript. There are many typo and grammar errors (e.g., lines 40, 96-100, 123, 135).
  8. Typesetting of Figures and Tables seem to be not good. Please rearrange them.
  9. Overall, the discussion section is poorly described. Please rewrite and reinforce extensively.  

Author Response

(Q) The Introduction section: please provide more information on nanoparticle albumin-bound (nab) with recently published references.

<Reply>

Thank you very much for your helpful comments. We added some information on nab technology with recent references in the revised manuscript (page2). 

(Q) Section 2.3. Preparation of HSA-DOX: please rewrite all the section since it was unclear. Please describe the lyophilization process in detail.

<Reply>

We apologize for any misunderstanding this may have caused. We rewrote all the section2.3., and added the method of the lyophilization process (page4).

(Q) There was no statistical analysis. Please include in the revised manuscript.

<Reply>

We appreciate your constructive suggestion. We added the method of statistical analysis in the revised manuscript (page 5).

(Q) Section 3.2. In vitro release profile of DOX from HSA-d-DOX (Fig 2): the release of free-DOX in two pH conditions should be determined for the comparison with the formulation group.

<Reply>

Thank you very much for your kind suggestion. We performed an additional in vitro experiment using HSA-m-DOX. The release profile of HSA-m-DOX was very similar to HSA-d-DOX. The result was added in Figure 2 and the result section of the revised manuscript (page6-7).

(Q) The x-axis of the Figure 3A should be log-scaled.

<Reply>

We changed to log-scale in the x-axis of Figure 3A.

(Q) In the Figure 4B, the reviewer found that the liver tissue has higher accumulation and HSA-d-DOX seems to have the highest values compared to control and HSA-m-DOX group. Please discuss this result somewhere in terms of active targeting. The high accumulation of the liver does not indicate the active targeting of the formulation. Please clarify it.

<Reply>

Previously, our collaborators demonstrated that modifications of basic amino acids on the surface of HSA increased 30-fold higher liver uptake of the HSA [Biochim Biophys Acta. 2006 Apr;1764(4):743-9., Biochim Biophys Acta. 2003 Oct 13;1623(2-3):88-97.]. In general, modifications of basic amino acids led to the blockage of the positive charges of HSA. To conjugate DOX to HSA in this study, 2-iminothiolane reacted with lysine residues on the surface of HSA. This reaction induced the blockage of the positive charges. Therefore, this evidence suggested that this 2-iminothiolane reaction might induce the increase of liver uptake of HSA-m-DOX and HSA-d-DOX. To avoid this problem, the other method of DOX conjugation without modification of basic amino acids would be necessary. This discussion was added to the revised manuscript (page8).   

(Q) What is the loading efficiency of the DOX for HSA-m-DOX and HSA-d-DOX. Is there any advantage of HSA-d-DOX?

<Reply>

We appreciate your important question. However, Table1 showed that the loading efficiency of the DOX for HSA-m-DOX and HSA-d-DOX is 4.08 mol/mol HSA-m and 7.9 mol/mol HSA-d, respectively. HSA-d comprises two albumin molecules. Therefore, the DOX loading efficiency for HSA-d-DOX is comparable to the HSA-m-DOX, so HSA-d-DOX has no advantage regarding the DOX loading efficiency. Our data showed that the only advantage of HSA-d-DOX is higher tumour distribution, resulting in that HSA-d-DOX possessed higher antitumour activity. So, we concluded that HSA-d could potentially be a useful drug carrier for the antitumour drug delivery system against human pancreatic tumours. We would appreciate your understanding.

Minor points:

(Q) Title: please consider avoiding abbreviation (HAS).

<Reply>

We had replaced 'HSA' with 'Human serum albumin' in our title.

(Q) Please define all the abbreviations before use (e.g., DDS, SUIT2 in line 74).

<Reply>

We had corrected them.

(Q) Please correct the units in lines 105, 113, and 133.

<Reply>

We had corrected them.

(Q) Lines 150-151: HSA-m-DOX (8.0 mg (DOX)/kg) was repeated. Please check again.

<Reply>

We apologize for any misunderstanding this may have caused. We have changed ‘HSA-m-DOX (8.0 mg (DOX)/kg)’to ‘HSA-d-DOX (8.0 mg (DOX)/kg)’(page5).

(Q) Table 2: *P < 0.05 vs. HSA-m-DOX. Free-DOX and HSA-m-DOX were significantly different but there was no significant symbol.

<Reply>

Thank you very much for your kind suggestion. We have added **P < 0.01 significant symbol between Free-DOX and HSA-m-DOX in revised Table 2.

(Q) Lines 349-352: please check.

<Reply>

We apologize for any misunderstanding this may have caused. These template's sentences were deleted in the revised manuscript.

(Q) Please carefully check the English usage for the whole manuscript. There are many typo and grammar errors (e.g., lines 40, 96-100, 123, 135).

<Reply>

Thank you very much for your kind suggestion. We have modified some sentences in the revised manuscript.

(Q) Typesetting of Figures and Tables seem to be not good. Please rearrange them.

<Reply>

We had re-attached all tables and figures.

(Q) Overall, the discussion section is poorly described. Please rewrite and reinforce extensively.  

<Reply>

Thank you very much for your constructive comment. We have added some information in the discussion section (page5-10).

Reviewer 3 Report

This manuscript has merit, but it needs mandatory revisions before publication, as detailed below.

  1. Abstract: The sentence “In conclusion, HSA-d 30 has the potential as a next-generation nab-technology for antitumour drug delivery system.” must be removed as it is highly speculative (and not supported by only this very limited study).
  2. The reference list is poor and needs to be enriched. The Introduction section must include other reported nano-DDS for DOX and also other published studies that use HSA-based systems as drug carriers.
  3. Section 3.1. must be deeply revised with a detailed description of the reaction conditions and detailed synthesis steps that allow other authors to reproduce the work.
  4. The release assays must be prolonged to, at least, 48 hours (release is not complete) and these assays must be compared with other published release assays for DOX.
  5. The comparison with only INNO-206 is not enough to support the claims.
  6. Mistakes to be corrected:

Line 123:  1 × 104 cells/well

Line 129:  1 × 105 cells/well)

Line 148: mm3

Lines 349-352: Conclusions: “This section may be divided by subheadings. It should provide a concise and precise de-scription of the experimental results, their interpretation, as well as the experimental conclusions that can be drawn.” should, of course, be removed.

The manuscript has several formatting problems, namely in figures and tables.

Author Response

(Q) Abstract: The sentence "In conclusion, HSA-d 30 has the potential as a next-generation nab-technology for antitumour drug delivery system." must be removed as it is highly speculative (and not supported by only this very limited study).

<Reply>

We have removed the sentence from the abstract and changed it to 'In conclusion, HSA-d has the potential as a useful drug carrier for antitumour drug delivery system against human pancreatic tumour.' in the revised manuscript (page1).

(Q) The reference list is poor and needs to be enriched. The Introduction section must include other reported nano-DDS for DOX and also other published studies that use HSA-based systems as drug carriers.

<Reply>

We appreciate the reviewer's constructive suggestion. We added some information regarding nano-DDS and other recent studies with references in the revised manuscript (page2 and reference list).

(Q) Section 3.1. must be deeply revised with a detailed description of the reaction conditions and detailed synthesis steps that allow other authors to reproduce the work.

<Reply>

Thank you very much for your kind suggestion. According to the reviewer comment, we add the detailed methods for the synthesis of HSA and HSA-DOX in the revised manuscript (page3-4)

(Q) The release assays must be prolonged to, at least, 48 hours (release is not complete) and these assays must be compared with other published release assays for DOX.

<Reply>

We performed an additional in vitro stability experiment using HSA-d-DOX for 48 h. The total release from HSA-d-DOX in pH 5.5 buffer at 48 h was around 80%, suggesting that not so different compared with other published release assays for DOX (ex. Drug Des Devel Ther. 2015 Sep 7;9:5123-33.). The result and related references were added in the revised manuscript (page6-7).

(Q) The comparison with only INNO-206 is not enough to support the claims.

<Reply>

Thank you very much for your kind suggestion. Some papers have already clarified that INNO-206 could significantly inhibit tumour growth compared with free DOX in vivo experiments [Small. 2019 Mar; 15(12): e1804452., Expert Opin Investig Drugs. 2007 Jun;16(6):855-66.]. Therefore, we decided to perform the in vivo experiment without the free DOX group and to compare it to INNO-206 group. We would appreciate your understanding.

(Q) Mistakes to be corrected:

Line 123:  1 × 104 cells/well

Line 129:  1 × 105 cells/well)

Line 148: mm3

Lines 349-352: Conclusions: "This section may be divided by subheadings. It should provide a concise and precise de-scription of the experimental results, their interpretation, as well as the experimental conclusions that can be drawn." should, of course, be removed.

The manuscript has several formatting problems, namely in figures and tables.

<Reply>

We had corrected them.

Round 2

Reviewer 1 Report

The authors thoroughly improved their manuscript and I do not have any major objections concerning this manuscript anymore.

I would recommend using Sans-Serif Font in the figures which improves the readablity. However, this is really a minor point and I think the authors did a good job even though the discussion can go a bit deeper but in the end this contribution seems to fit the journal requirements for "Pharmaceutics".

Author Response

Thank you very much for your positive comments. We have used Sans-Serif font to replace the original font in the figures.

Reviewer 2 Report

The authors tries to reflect the reviewer's suggestion to some extent. However, there are still minor points to be improved as below.

  1. As the reviewer suggested the extensive rewriting and reinforcement on discussion section previously, the present discussion remains still weakness in the revised manuscript.  For examples, HSA-d-Dox showed best tumor accumulation and anticancer effect in vivo than HSA-m-DOX and INNO-206, whereas the anticancer activity (i.e., cytotoxicity: IC50) and cellular uptake of HSA-d-DOX  is lower than Free-Dox. In the introuction the INNO-206 has similar anticancer activity compared to free Dox (line 84-90), so that the present in vitro anticancer activity (i.e., cytotoxicity) seems to be inconsitent with in vivo observation. This discrepancy should be discussed somewhere. 
  2. In the introuction the INNO-206 has similar anticancer activity compared to free Dox (line 84-90). However, the antitumor activity of INNO-206 did not show significant effect than PBS treated group, suggesting lack of anticancer activity of INNO-206 in vivo. Please discuss and compared the present data with literature data of INNO-206 more.
  3. What is a unit of Y-axis in Fig3B?
  4. Diameter and DOX/HSAs (molar ratio) are presented repeatedly in Table1 and Figure1B and 1C. Please clarify it.

Author Response

The authors tried to reflect the reviewer's suggestion to some extent. However, there are still minor points to be improved as below.

  1. As the reviewer suggested the extensive rewriting and reinforcement on discussion section previously, the present discussion remains still weakness in the revised manuscript.  For examples, HSA-d-Dox showed best tumor accumulation and anticancer effect in vivo than HSA-m-DOX and INNO-206, whereas the anticancer activity (i.e., cytotoxicity: IC50) and cellular uptake of HSA-d-DOX is lower than Free-Dox. In the introduction the INNO-206 has similar anticancer activity compared to free Dox (line 84-90), so that the present in vitro anticancer activity (i.e., cytotoxicity) seems to be inconsistent with in vivo observation. This discrepancy should be discussed somewhere. 

<Reply>

Thank you for pointing out the discrepancy in the introduction. We apologise for our oversight. We have revised the respective sentence to:

Administered at the equivalent dose, INNO-206 is safer than free DOX due to its improved kinetics and antitumour activity than free DOX [9].

In addition, we have added the discussion about the difference of anticancer effect between in vitro and in vivo in the revised manuscript:

Although Table2 showed that free DOX possesses higher cytotoxicity than HSA-d-DOX in vitro, HSA-d-DOX possesses the highest antitumour activity among other drugs. These data strongly suggest that the tumour accumulation and retention of HSA-d-DOX are very superior to other drugs.  

  1. In the introduction the INNO-206 has similar anticancer activity compared to free Dox (line 84-90). However, the antitumor activity of INNO-206 did not show significant effect than PBS treated group, suggesting lack of anticancer activity of INNO-206 in vivo. Please discuss and compared the present data with literature data of INNO-206 more.

<Reply>

We have added the following paragraph in the discussion:

The following dosing rates have been reported to be effective: Day10, 17, 24 dosings of DOX 24 mg/kg [Invest New Drugs. 2010 Feb;28(1):14-9.], and Day 10, 17, 24 dosings of DOX 12 mg/kg [Int J Pharm. 2013 Jan 30;441(1-2):499-506.], while the dosing rate used in the present study, Day 0, 7 dosings of DOX 8.0 mg/kg, practically showed no antitumor efficacy. This 2 x DOX 8.0 mg/kg dosing rate is why the antitumor activity of INNO-206 did not show a significant difference from that of the PBS treated group. In contrast, despite dosing at 2 x DOX 8.0 mg/kg, HSA-d-DOX showed antitumour activity.      

  1. What is a unit of Y-axis in Fig3B?

<Reply>

MFI refers to Mean Fluorescence Intensity, and we have rewritten MFI in full. 

  1. Diameter and DOX/HSAs (molar ratio) are presented repeatedly in Table1 and Figure1B and 1C. Please clarify it.

<Reply>

Figure1 shows the preparation scheme of HSA-d-DOX, and the stability of HSA-m-DOX and HSA-d-DOX particles before and after lyophilization. On the other hand, Table1 shows the physicochemical property information of both HSA-m-DOX and HSA-d-DOX.

We have revised the titles of Table 1 and Figure 1 as below:

Table 1 Physicochemical characterization of HSA-m-DOX and HSA-d-DOX.

Figure 1 The preparation scheme of HSA-d-DOX, and the stability of HSA-m-DOX and HSA-d-DOX particles before and after lyophilization.

Reviewer 3 Report

The authors have improved the manuscript according to the reviewers'  suggestions.

Author Response

Thank you very much for your comment.